# A Review of Age Friendly Virtual Assistive Technologies and their Effect on Daily Living for Carers and Dependent Adults

**DOI:** 10.3390/healthcare7010049

**Published:** 2019-03-21

**Authors:** Hannah Ramsden Marston, Julie Samuels

**Affiliations:** 1Health & Wellbeing Priority Research Area, School of Health, Wellbeing & Social Care, The Open University, Milton Keynes, Buckinghamshire MK7 6HH, UK; 2Independent Researcher, Lincolnshire 60069, UK; samuelsjulie@hotmail.com

**Keywords:** ICT, Age in Place, Disability, Smart Technology, Intergenerational Relationships, Connected Health

## Abstract

Many barriers exist in the lives of older adult’s, including health, transport, housing, isolation, disability and access to technology. The appropriate integration of technology within age-friendly communities continues to offer possible solutions to these barriers and challenges. Older adults and disabled people continue to be affected and marginalized due to lack of access to the digital world. Working collaboratively with planners, policy makers and developers, social and living spaces in the future will ensure that residents are equipped to live in an era that continues to be led by, and is dependent upon, access to technology. This review paper uniquely draws together the small volume of literature from the fields of gerontology, gerontechnology, human computer interaction (HCI), and disability. This paper examines the national and international age-friendly frameworks regarding older adults who are carers of dependent people with disabilities.

## 1. Introduction

The age friendly movement commenced in 2007 when the World Health Organization (WHO) set out its global plan and framework [1] for Age-Friendly Cities. The WHO defines age-friendly as “*policies, services, setting and structure support and able people to age actively*” [1] (p. 5). This programme brought together 33 cities across 22 countries to identify and ascertain what key elements within the urban environment facilitated and supported active and healthy ageing (AHA) [2]. The WHO Global Network of Age-Friendly Cities was established for four reasons:(1)To link participating cities to WHO and to each other;(2)To facilitate the exchange of information and best practices;(3)To foster interventions that are appropriate, sustainable and cost-effective for improving the lives of older people; and(4)To provide technical support and training [2].

To date, the global population stands at nearly 7.6 billion people, with 60% of world’s population residing in Asia, 17% in Africa, 10% in Europe, and 9% in Latin America and the Caribbean. China and India continue to be the most populous countries, with 19 and 18% respectively [3]. The growth of the population is increasing at a rate of 1.10 per cent per year, slower than the last decade at 1.24 per cent per year. By 2030, the United Nations (UN) estimate the global population will reach 8.6 billion and increase to 9.8 billion by 2050. The global population is estimated to rise to 11.2 billon people by 2100 [3]. 

The WHO Global Network of Age-Friendly Cities builds on the WHO active ageing framework. Fitzgerald and Caro [4] reported the WHO definition of ‘active ageing’ as “*the process of optimizing opportunities for health, participation, and security in order to enhance quality of life as people age. It applies to both individuals and population groups*” (p. 12) [5]. More recently, the WHO has replaced the previous policy [5] with the term and notion of ‘Healthy Ageing’ [6]. The WHO ‘Healthy Ageing’ policy has been set as a goal to achieve between 2015 and 2030 [6]. The WHO defines Healthy Ageing as “*the process of developing and maintaining the functional ability that enables wellbeing in older age*” [6]. Under this policy, the needs and abilities of an individual are measured through the following criteria:Meet their basic needs,To learn, grow and make decisions,To be mobile,To build and maintain relationships, andTo contribute to society [6].

There are several points which one should consider under this definition, including: the *intrinsic capacity* associated with the mental and physical abilities of an individual (i.e., walking, thinking, seeing and hearing). These can be affected by a disease, injury and age-related conditions. *Environment* includes several factors, including the home, community and society, interwoven in conjunction with relationships, attitudes, values and policies. Health and social care provisions and systems should ideally be interconnected, in a bid to support individuals’ *intrinsic capacity* [6]. There are two primary considerations noted by the WHO and their Healthy Ageing framework:(1)**Diversity:** There is no typical older person. Some 80-year olds have levels of physical and mental capacity that compare favourably with 30-year olds. Others of the same age may require extensive care and support for basic activities like dressing and eating. Policy should be framed to improve the functional ability of all older people, whether they are robust, care-dependent or in between [6].(2)**Inequity:** A large proportion (approximately 75%) of the diversity in capacity and circumstance observed in older age is the result of the cumulative impact of advantage and disadvantage across people’s lives. Importantly, the relationships we have with our environments are shaped by factors such as the family we were born into, our sex, our ethnicity, level of education and financial resources [6].

This paper examines the national and international age-friendly frameworks with respect to older adults who are carers for people with disabilities. Within its overview of existing age-friendly frameworks and contemporary evidence, an overview of state-of-the-art technologies is presented, followed by recommendations for expanding this work. 

## 2. Background Literature

### 2.1. Age-Friendly Communities

Since the turn of the millennium, there have been several age-friendly initiatives building on the WHO Global Age-friendly framework [1]. A review conducted by Steels [7] provides a synopsis of global age-friendly cities and frameworks [2,8,9,10,11,12], illustrating their key features. Fitzgerald and Caro [4] presented several features and elements which they deemed necessary to meet the minimum requirements of an age-friendly city or community. These features include pre-conditions that must be in place before any age-friendly initiative can commence. The preconditions include: the density of population, climate, weather, topographical features (communities residing on hills such as the favelas in Brazil), social and civic organisations, health and social care provision [4]. Within diverse communities, ensuring residents have a variety of mobility options is crucial. This includes ensuring the availability of public transport connections, accessible places to walk, and community transport services (e.g., dial a ride). Within these communities, the requirements of outdoor spaces and buildings to facilitate and enable residents to successfully age in place is central to the success of the environment. Early consultations between residents, planners and developers to identify key challenges and potential barriers are crucial to the achievement of all of the above aims. Barriers and challenges may not be identified, challenged and amended without consultations between the residents, planners and developers. Moreover, community activities require that residents respectfully build up their relationships with one another in conjunction with their friends, family and support networks, ensuring residents are respectful of each other [4]. Several approaches such as focus groups, face-to-face meetings, interviews (e.g., one-to-one or community) or surveys can be conducted to identify needs and requirements from residents. It is important to ensure that all interested parties have the opportunity to communicate and share their expectations and concerns. 

Buffel [13] has conducted co-researching and co-production activities as part of the Manchester Age-Friendly strategy (MAS) [2]. A significant element in the success of MAS is the enablement of actors to share their experiences and learn from each other about the needs and requirements of their communities. The concept of an ‘Age-Friendly Business’ has enabled businesses in the community to make alterations to facilitate ease of access or service by residents [4]. Such alterations or changes include: assisted devices to open doors, increasing the font size on menus, and changing the height and access to toilet dispensers [4]. Already, across America and Ireland businesses have undertaken alterations to provide residents with ease of access to the premises or services. Consequently, many businesses increased their income as a result of word-of-mouth approval in the community [4]. 

In the UK, the city of Manchester was recognized by the WHO as an age-friendly city-region, chosen as part of the age-friendly initiative programme [14]. McGarry and colleagues [2] provide an overview of the two frameworks and approaches set out by the WHO and MAS [15]. 

The WHO age-friendly strategy includes eight domains of interest: (1) Outdoor spaces and buildings; (2) housing; (3) transportation; (4) social participation; (5) respect and social inclusion; (6) civic participation and employment; (7) communications and information and (8) community support and health services. Primarily, the MAS 2020 age-friendly strategy [15] focuses on six of the eight domains outlined by the WHO and includes: (1) lifetime neighbourhoods (environment, community safety, housing, transport); (2) cross-cutting themes: improving engagement, improving relationships; (3) cross-cutting themes: promoting equality; (4) income and employment; (5) culture and learning, and (6) healthy ageing, care and support services [2]. 

To date, age-friendly initiatives have primarily focused on the needs and requirements of existing ageing populations. However, there is little consideration and discussion surrounding the needs of mid-older adults (<~45 years old) who are carers of children/young people and dependent adults with disabilities. Furthermore, what are the implications, based on the national and international age-friendly strategies associated to successfully age in place? 

The aim of this paper is to review age-friendly virtual assistants and their effect on carers and dependent adults in contemporary society. 

### 2.2. Methods

This paper will be underpinned by identity theory posited by Burke & Stets [16]. The notion of identity theory posits persons residing in society the opportunity to reserve a stable environment, irrespective of any slight inconsistencies. This is succeeded by the change in peoples’ actions, which in turn results in the perceptions of persons aligned with the standard or ideal self [16], while the balance within one’s environment shifts based on the deviation or non-verification of a person’s identity; this results in a person’s modification of their behaviour. 

For those parents, guardians and carers who are residing in disability friendly communities and age-friendly environments, they have the opportunity to continue to live alongside their dependent child. This, in turn, has the potential to alleviate stress, social isolation, loneliness, and promote independence for both carers and dependent adults. Giving dependent adults the opportunity to live semi-independently or fully-independently alongside their parents, guardians and carers will contribute to the creation of and dissemination of good practice. 

Age-friendly cities can provide harmonious, supportive, inclusive living and social environments regardless of age, race, gender or disability through face-to-face communications, created and supported by technology. Through a myriad of activities, age-friendly cities can help to identify, and tackle problems experienced by the residents, while providing immediate and on-going support for both the carer and the dependent adult. Conversely, by encouraging the opportunities for social encounters and by building upon the existing age-friendly frameworks, there are potential benefits and improved social cohesion for all residents in the wider community. 

### 2.3. Digital Exclusion

Automation and accessibility of goods and services (e.g., banking, shopping, health care appointments) are increasing, which is resulting in limited access by some populations. According to the UK Government “*Digital inclusion, or rather, reducing digital exclusion, is about making sure that people have the capability to use the internet to do things that benefit them day to day*” [sic] [17]. Moreover, in the UK Government Digital Inclusion Strategy 2014 policy paper, Francis Maude, Minister for the Cabinet Office, stated, “*We need to equip the whole country with the skills, motivation and trust to go online, be digitally capable and to make the most of the internet*” [17]. 

A worrying phenomenon relates to the continuation of digital exclusion relating to the “vulnerable and disadvantaged groups in society” [17]. The policy identified five groups within society that are most likely to be digitally excluded: (1)Those in social housing,(2)Those on lower wages, or unemployed,(3)Those with disabilities,(4)Older people, and(5)Young people. Only 27% of young people who are offline are in full-time employment [17].

Within existing debates surrounding digital cities, there are vulnerable members of society who are marginalized and penalized because of limited access to and understanding of the digital world. This in turn, has the potential to be a detrimental factor relating to their health and independence. 

The digital divide is still an ongoing topic of discussion, which results in many communities, and individuals not being able to access rudimentary technologies such as a computer and/or access the Internet [18,19,20,21,22]. Ferguson and Damodoran [23] reported how the digital divide primarily focuses on the ‘haves’ and the ‘have nots.’ However, it also relates to three points that differentiate those who associated with the digital divide. First, connectivity: this relates to appropriate access to equipment. Second, capability: ensuring everyone has the skills and knowledge to conduct tasks and to retrieve relevant information. Finally, content: the perception of relevant content and the *“motivation from the ‘pull’ of compelling functionality”* (p. 5) [23]. 

Within the digital divide, digital participation is important for all citizens, be it those who are vulnerable, or who are slow adopters, from the older/elderly person to the wealthiest of individuals. It is important to understand the motivations of digital participation to understand what the barriers are to using technology, and to ensure access and availability is met. Ferguson and Damodoran [24] stated:

“[…] *widespread digital participation can only come about through the confident and successful take up by older people and others in the digital world and the way that services relevant to their needs are designed and presented*”.(p. 5)

The authors believe that by collaborating and communicating with communities at both grassroots networks and with national organisations, there are opportunities to learn and understand the barriers to and enablers of technology faced by older adults and slow adopters. Currently, there is a growth of work in the domains of the digital divide and older adults’ engagement. However, there is a paucity of work surrounding those individuals known as ‘slow adopters.’ This cohort of society are individuals or communities who are not just older people, but who are mid-older and younger people, who may live in social housing; they are people who are unemployed or who are employed on precarious contracts or receive low incomes. They are individuals with disabilities, who reside in different communities both culturally (e.g., traveller) and geographically (e.g., rural), or who are homeless and are moving between towns and cities, or who are moving around different areas within one place, or young people who are not in employment, education or training (NEETs). All of these categories of citizens may have no direct or limited access to public funds [24]. For some citizens, their level of literacy, numeracy and digital literacy skills pose additional barriers to their digital participation. 

In January 2019, the NHS (National Health Service) Digital announced the commencement of a project focusing on the use of digital technologies by homeless community outreach workers in Hastings, UK [25]. The project collaborates with several partners including NHS England, NHS Digital, Good Things Foundation and The Seaview project. The aim is to use digital technologies to ensure a suitable approach is conducted by and between support workers/organisations and the homeless community. Physical locations such as libraries can provide public access to rough sleepers who wish to search for specific information (e.g., health and wellbeing centres) on the Internet. However, little work has focused on the barriers and enablers to technology faced by individuals of the homeless communities. Additional issues and challenges can hinder technology use by rough sleepers and include the varying types of data plans, access to charging points for a mobile/smartphone, and possible exclusion from accessing public Wi-Fi. Furthermore, many mobile phone plans require a data contract and bank account, which too could be problematic for rough sleepers who do not have access to this type of information or accounts. There are many reasons why slow adopters and older adults have barriers to adopting and using technology in their lives. This can include embarrassment around their lack of technical knowledge and skills, and the design of technologies, while there could be limited opportunities for learning outside of the workplace [24]. By understanding the needs of marginalized and disadvantaged communities, support and guidance can be offered to ensure individuals within these respective communities can become digital citizens. The UK Government [17] has outlined its digital strategy, although, when it was outlined three years earlier, Adam Hillmore stated:

*“We should not consider increasing online presence among older people on its own; it is easier to bring people together as a community and to make using the internet part of that”*. (p. 5) [24]

Taking a grassroots approach, as suggested in the quote above should safeguard all voices are heard within respective communities. Ferguson and Damodoran noted that the position of local governments is ideally situated within their communities to take the lead and to facilitate a ‘user-driven’ approach [24]. Local government is the ideal actor to take the lead role within communities and towns regarding digital participation. Given that local governments own public space and buildings such as libraries, they are ideally placed to input into schools and partnerships. Taking on this role, local governments can encourage their respective networks and partners to become active members across their communities. This, in turn, may link to different initiatives that can also benefit from local government assisting with key issues [26]. Furthermore, understanding the exact needs and requirements of marginalized communities is facilitated by employing a co-design/creation approach rather than a top-down process. Seven needs have been reported by Ferguson and Damodoran [23] based on specific user characteristics:(1)Readily available,(2)Trusted and sustained,(3)Delivered in familiar, welcoming and local venues,(4)Embedded in social activities and personal interests,(5)Free of time pressure and assessments,(6)Inclusive of problem-solving/trouble-shooting, and(7)Offering impartial advices and ‘try before you buy’ [23,27].

Furthermore, Ferguson and Damodoran argued:

*“The UK Government Digital Strategy (launched 1 March 2017) states that it seeks to simultaneously implement strategies intended to address connectivity issues (with the aim of completing the roll-out of 4G and superfast broadband by 2020) and capability issues (e.g., creating the Digital Training and Support Framework)”*.(p. 6) [26]

Nonetheless, there are concerns that still need to be addressed for rural and marginalized communities, and for those individuals who are slow adopters and older adults. The latter is equally important, because, for many people, they learn how to use technology in the workplace. For those people who have retired and were not exposed to technology, this too will result in limited and low digital participation [26]. Moreover, Ferguson and Damodoran [26], note how evidence indicates *“basic skills training has reached most of those for whom it is appropriate”* (p. 6). However, while basic skills training may benefit some people in society, for those who are slow adopters, it is likely that they have not had the opportunity [26]. 

Indeed, for many people using technologies to access the digital world, it is an integral part of their daily lives and “*not using the internet is different from ‘digital exclusion.’ Some non-users have made an informed and reasoned choice to be offline*” (p. 1) [28]. While access to the digital world is available in both public and social spaces, one requires a digital device (i.e., a computer, laptop, smartphone or tablet) to access the Internet, which in turn leads onto other digital worlds. Moreover, the Centre for Ageing Better notes, “*As opposed to digital inclusion operating as a standalone intervention, digital support should be embedded within the delivery model of a range of local community and public services wherever feasible and appropriate*” [28]. With on-going austerity and cuts to public services, it is of paramount importance that local and national governments do not marginalize and penalize vulnerable members of society further. 

Mouland, Richardson and Damodoran [28] stated “*Even for those who are engaged with existing technologies, the pace at which technology develops places significant demands on us to learn new behaviors and skills. Those who were raised in a digital world will still hit these obstacles over time and find new technologies harder to adopt—particularly after leaving the workforce*” (p. 6) [28]. 

## 3. Technology Solutions

Phenomenal technology developments have occurred over the last twenty years in the field of enabling industries (i.e., video games, smartphone and small, medium enterprises (SME)) the opportunity to design, develop and enhance solutions to reach a broad spectrum of users in society. 

In the proceeding section, the authors will review different technologies, ICTs and contemporary research projects aimed to facilitate and enhance users’ accessibility and ease of use, to support successful ageing in place through active and healthy ageing (AHA). 

### 3.1. Overview of Virtual Assistants

In recent years, we have seen the development of what is been coined as ‘personal assistants’ or ‘virtual assistants,’ designed in the form of ‘speakers’ that can be placed around the house and respond to a voice(s), which in turn executes the command(s). The most commonly known devices are smart speakers/personal assistants such as the Amazon Echo or Alexa [29,30]. There are other devices with similar capabilities known as Google Home and Google Home Mini. Contemporary research has suggested these ‘personal assistants’ can offer older adults the ability to maintain living independently, and possibly support ageing in place. Indeed, according to the National Institute for Health Research (NIHR), “*A number of studies have explored integrated monitoring and response systems to check the health, wellbeing and safety of older people living at home. Some of these are focused on particular groups, like those with dementia. They range from systems using sensors, alarms or wearable technology to cameras, smart televisions and service robots*” [31] (p. 3).

Therefore, the functionality of these virtual or personal assistants can provide a user with a wealth of information (e.g., weather reports, checking events in a calendar), coupled with the ability to control their heating and lighting on or off via three automation and third-party apps. 

Additional features available through the virtual assistants offer users the opportunity to control what music they listen to (via streaming services), set and manage alarms, order food (e.g., Domino’s or Pizza Hut) and set reminders (e.g., for medication) [17,18,19,20,21,22,23,32,33,34,35,36,37,38,39,40,41]. Homes which have ‘home automation’ virtual assistants have the capacity to interact and connect with several manufacturers, including Philips Hue and Nest [39,40,41]. 

Some existing users of Alexa say they feel a strong bond with their virtual assistant and perceive their devices as a member of the family [30,41]. The notion of using Alexa and similar devices or virtual assistants within the home can offer the users or residents a multitude of opportunities to engage and receive information. Whether you are an older adult, a carer or a dependent adult, there are opportunities to age in place by connecting with these types of devices through the primary interaction of voice recognition. 

Early adoption of new technology is key. For many people in society, learning how to use a new piece of technology can be worrisome or a steep learning curve [42]. Nevertheless, for some people it is crucial that the technical infrastructure allows several devices to seamlessly operate together, in order to deliver an automated, self-monitoring smart home [43,44,45,46].

Li and colleagues [44] proposed the notion of neighbourhoods being connected via wireless sensors, which, if triggered through deviant activity, can be recorded via surveillance cameras, which in turn would inform all residents connected on the smart system. This follows the original conception of the ‘Neighbourhood Watch’ scheme across the UK, where residents involved in the scheme reported any suspicious behaviour or crime to the police. A ‘smart, connected,’ age-friendly and disabled-friendly community gives residents the potential to detect problems and protect one another.

### 3.2. Integrating Virtual Assistants into the Lives of Carers and People with Disabilities

While research concerning virtual assistants is still in its infancy, these devices have great potential for people with a multitude of disabilities. For example, Hampshire County Council trialled the Amazon Echo to help both the elderly and disabled [47,48] people in their communities. Similarly, Virgin Trains have integrated Alexa to assist disabled passengers [49] with their communication and interactions. Several factors affect dependent adults (e.g., physical, cognitive, speech and visual impairments). These include low self-esteem and confidence and limited social networks, which in turn increase their risk of loneliness, poor health and wellbeing. In some instances, one’s disability may fluctuate throughout one’s life, and may deteriorate over time. Therefore, the use of virtual assistants (Amazon Echo, Alexa, Google Home and Google Home mini) can offer dependent adult’s additional options to communicating with friends and family members, more so than only participating in dialogue when a specific answer is needed. 

Devices including Amazon Echo, Alexa, Google Home and Google Home mini provide individuals with disabilities the opportunity to communicate with the device and respond to commands. One example of a disability that may affect an individual’s communication (i.e., speech and language) is autism spectrum disorder (ASD). Many individuals who have been diagnosed with ASD require an intervention focusing on the “*aspects involved in producing or understanding speech and language*” [50]. The use of these devices may initially cause frustration for the individual, given the initial inability of the device(s) to decipher speech. However, one of the benefits of virtual assistants is the potential to help improve speech. 

An example of this is demonstrated in the written account of Megan D, who reflected on her six-year-old, disabled son, and who used the virtual assistant Alexa. Megan D noted how her son communicates and connects with people through a series of questions that fall into his areas of interest. Regardless of whether the son has asked these questions earlier on in the day or in previous days, the use of Alexa is the primary way for the son to connect and communicate [51]. Since purchasing the Alexa device, Megan D has observed changes in her son. Typically, through repetition of questions, Megan D’s son regularly engages in conversations with the Alexa device. However, it is the virtual assistant’s capacity to answer the son’s repetitive questions on demand that has improved the quality of life of Megan D and her son. Furthermore, the ability to recognize different voices and language patterns makes this type of virtual assistant an ideal companion for many individuals with ASD. 

The use of virtual assistants can be beneficial to individuals who have been diagnosed with a neuro-degenerative condition such as multiple sclerosis (MS). Hampshire County Council, in conjunction with NHS Digital, have launched a scheme which explores the integration of the Alexa device into adult service care plans. For example, Claire Williams, who has been diagnosed with MS, was one of the recipients of these virtual assistants. Williams notes “*I can do loads of things for myself now which when I was first diagnosed with MS seven years ago, I didn’t think I’d be able to do.*” Furthermore, Williams has reported positive improvements within her life due to the integration and use of the Alexa virtual assistant. Williams has used the device for many things, including turning on the lights, playing music, reading books and adding items to the shopping list, which is on her husband’s phone [52].

Likewise, Bogost shares his experiences of using Alexa with his 82-year-old father, who is legally blind and has been since the age of 18. Bogost notes how the virtual assistant offers a “*h**ands-free operation for able-bodied folk and new accessibility for those with limited mobility or dexterity*” [53]. Furthermore, Bogost [53] recognises his father’s willingness to embrace the new technology, for example, using it playfully by asking the virtual assistant a series of questions he knows it cannot answer.

While we have focused on the use and installation of virtual assistants in the home and an age-friendly environment, there are further opportunities for bringing ageing and disability together. These opportunities do not just safeguard against social disconnectedness, reduce isolation, and improve communication, but also ensure physical and cognitive fitness can be maintained, therefore demonstrating how ICT can help an AHA be achieved. There have been several key pieces of research that have integrated technology investigating both physical and cognitive activities. In the following section, we provide an overview of contemporary research that has the potential to offer older adults, carers and dependent adults the opportunity to engage and interact with the technologies aimed at enhancing AHA and intergenerational relationships.

### 3.3. Overview of Exergames

The iStoppFalls (ISF) European Union (EU) Project [54] was an international, multi-centre study, which included a single-blind, two-group randomized control trial (RCT) involving 160 community-dwelling older adults aged 65 years and over [55]. ISF aimed to design and develop information, communication and technology (ICT) based systems using physical activity to reduce the risk of falling by adults aged 65 years and over. Additionally, strength and balance exercises were conducted via user interaction and engagement with three purpose-built exergames. Gschwind and colleagues [55,56,57] describe the ISF ICT-based system, which was comprised of several types of technologies (Figure 1 and Figure 2). The ISF system offered users a diverse range of interactive approaches (Figure 3), including gesture, a remote control, speech and a tablet device. Participants randomized to the intervention group (IG) had access to the ISF system through several menu options (Figure 4). These included: fitness training, reviewing user performance, meeting point (for example, virtual meeting place for all users), falls and health prevention. 

Three purpose-built exergames were designed, developed and implemented into the ISF system. Marston and colleagues [57,58] provide an extensive overview of the purpose-built exergames: the Bumble Bee Park, Hills ‘n’ Skills and The Bistro exergames (Figure 5). Each exergame incorporated strength and balance exercises from the Otago programme [58], while additional Otago exercises (Figure 6) were integrated into the system under the ‘training programme’ option. Therefore, users were able to continue building their strength and balance in conjunction with the exergames. 

Fall risk assessment was integrated into the ICT-based system, which enabled users to be initially assessed and included a physical assessment using the purpose-built software, the Microsoft Kinect console and the Senior Mobility Monitor (SMM) developed by Philips Netherlands [55,57]. Four physical assessments were conducted between the user and the integrated sensors which in turn enabled interaction via the user’s television. The assessments included several balance tests: comfortable-bipedal, semi-tandem, near-tandem and the tandem stance [55,56,57]. Participants were required to undertake the balance assessments twice for a maximum of 30 s each, leading with their preferred foot. Participants were instructed not to change their preference (foot) in between stances and to keep their eyes open. Reaction time was integrated and assessed in the ICT system through hand and foot reaction times for each respective participant. This was calculated by hitting the green button (when highlighted) on the table or on the floor of the virtual environment [55]. 

The ISF RCT concluded that the ICT-based system did reduce the physiological fall risks of older adults aged 65 years and over that were living in their own home. Participants assigned to the intervention group showed greater adherence and an improvement in postural sway, step reaction and executive function [56]. 

To understand the usability, user experience and acceptance of technologies within the ISF purpose-built system, Vaziri and colleagues [59] deployed the System Usability Scale (SUS) [60], the Physical Activity Enjoyment Scale (PACES) [61] and the Dynamic Acceptance Model for the Re-evaluation of Technologies (DART) [62], coupled with interviews and observations of participants. The results showed the ISF ICT-based system to have an overall score of 62 out of 100, indicating good usability, with most users enjoying the ISF exergames. The PACES measure and the DART measure displayed user acceptance of the ISF system to be acceptable. 

### 3.4. MobiAssist Project

The MobiAssist project (2015–2018) proceeded the iStoppFalls Project and aimed to explore the social impacts of the ICT-based suite of exergames aimed at people with dementia and their caregivers. Over a period of eight months, researchers used a co-design approach, while observing the daily lives of informal and professional caregivers of 14 people who had been diagnosed with dementia [63,64]. Conducting a co-design approach enabled the research team to gain insights into the daily routines of the participants’ and their caregivers, coupled with biographical backgrounds, memories, social environment and recording their experiences, attitudes and practices of using technology [64]. 

Participants were aged between 72–89.6 years, comprising six females and eight males [65]. The contents of the MobiAssist system include several digital technologies, software and purpose built exergames (Figure 6) [63,64]. The MobiAssist ICT-based system contains exercises and exergames enabling users’ performance to be measured and aims to “*counteract the progression of dementia and to help people with dementia to remain as autonomous as possible*” [64].

The MobiAssist project includes a series of strength and balance training exercises (Figure 7) from the Otago Exercise Program (OEP) [66], similar to the ISF ICT-based system [63,64]. The strength training exercises from the OEP enable users to strengthen their upper and lower limb muscles, using knee extensions, knee bends, sideways leg raises, toe-stands, the elbow bends and front raise aimed at the shoulder muscles [63,66]. The exergames aim to enhance and improve the balance and coordination of the participants [64]. Figure 8 left displays a visual representation of some of the games that are implemented into the MobiAssist project. 

A brief description of some of the games include: the ‘Apple game,’ which requires participants to collect virtual apples from a tree and place them into a basket; the ‘Mole game,’ which requires the participants to hit moles when they pop up from the ground at intermittent times. Participants’ engaging with the ‘Mole game,’ requires the user to move sideways and move forward (take steps) to hit the mole [64]. An additional game (Figure 8, left) is the ‘Wheel of Fortune,’ which requires the participants to raise their hands and spin the wheel. This game is aimed at problem solving and cognitive tasks such as letter games, mental arithmetic, classification and completion of rhymes, verses and poems or remembering music titles [64]. The second game, displayed in Figure 8, right, is based on folk music, and, while the participant marches on the spot, the music continues to play. However, if the participant stops marching on the spot, the music gradually fades out. If the participant wants to continue listening to the music, they have to restart marching on the spot [64]. 

MobiAssist project reported several limitations, including the different settings and system issues experienced by participants. For example, there were issues surrounding the Kinect recognition by participants, relatives or caregivers who were standing too close to the camera. There was an in/exclusion criterion to assist with the recruitment of participants and their support network(s). In order to ensure coherent recruitment, process a mediator (care institution) was involved in the recruitment procedure between the respective participants and the research team. Furthermore, conducting interviews with participants who had been diagnosed with dementia was difficult at times, in particular engaging with the MobiAssist ICT-based system. Therefore, participants were limited in their ability to provide the research team with “*meaningful and informative answers, largely because of the deterioration of their cognitive and communicative resources and capabilities*” [64]. 

Both the iStoppFalls and MobiAssist ICT-based systems have the potential to offer carers and dependent adults the opportunity to engage and socially connect with friends, family members and with each other. The technology infrastructure will be a key concern (e.g., cost of Internet connection) for some people, highlighting the very essence of the digital divide. The MobiAssist project shows that there is potential to enhance social interaction and increase empowerment using serious games, and simultaneously build intergenerational relationships between the carer and the individual. For example, the research team concluded that the MobiAssist system has a positive trend to “*support workflows and thus improve institutionalized quality of care*” [64]. 

Within an age-friendly environment, older carers and disabled people can connect and share experiences with one another. The technology solutions discussed in the previous section offer users across different age cohorts the motivation and opportunity to interact with both young and older cohorts. This, in turn, has the potential to reduce the risk of social isolation and enhance social connectedness, offering enhanced engagement, communication and ensuring the AHA mandate is achieved. 

Conversely, within the home environment and/or community, the notion of ageing-in-place can in some instances require assistance from younger adults of the family or community. Thus, integrating an intergenerational approach within an age-friendly environment has many positive benefits from the perspectives of both younger and older generations. These benefits include learning and sharing knowledge and experiences, caring opportunities for those who have fallen due to short, medium- and long-term illness, enhancing and build upon one’s social skills, and enhancing social connectedness, which in turn will result in a decrease of social isolation. Steels [7] examines the Generations of Hope Community (GHC) [65], a non-profit organisation and social welfare agency, which created a program located in Illinois, U.S., “*where children adopted from foster care can find permanent homes and develop intergenerational relationships in a specially designed community*” [67] (p. 48). 

Hope Meadows is a neighbourhood located two hours outside of Chicago. It became the first community planned by GHC and aimed “*to improve the service delivery and policies of the child welfare system; it ended up helping not only foster and adopted children but senior citizens as well.*” (p.18) [68]. 

Marc Freedman stated, “*The story of Hope Meadows offers not only a vision for how we can help take care of some of the most vulnerable young people in the society—foster children who essentially have nowhere else to turn—but how we can create neighborhoods that enrich the lives of all ages.*” (2001) [67].

At GHC, there are at least three generations residing in the environment—older adults, families and young people, facilitating a capacity to ensure care and support is available for the residents. By integrating an intergenerational approach into the environment, it can offer different generations of residents living within this type of environment an opportunity to undertake caring responsibilities, whereby the younger residents (e.g., children/teenagers) can understand the ethos of giving and receiving care in the future [68]. Through this social cohesion, each generation facilitates and teaches the others the different complexities, issues, needs and requirements which are significant to them, while learning from one another. 

The approach undertaken at GHC enables older adults “*who do not want retirement to mean the end of their productive years, who want it to mean something more than a pension, health care, and a roof over their heads*” [68] (p. 51).

This concept facilitates a myriad of individuals and families to live together in one community, serving a purpose for all residents. Utilizing the theoretical approach of identity theory purported by Burke and Stets [16] in the age-friendly home and framework, it offers the residents the opportunity to take on the role of carer. In particular, Hope Meadows facilitates older residents to re-establish identities and roles previously held in society and their respective communities. For some older residents in Hope Meadows, they have the opportunity to feel needed and/or useful through the eyes of the parents of young children as a knowledgeable friend or community member. In some instances, for the older residents, having the younger residents in the community who may need assistance or care will provide the older person a sense of purpose. This notion also offers co-residents the opportunity to share knowledge and experiences, thus resulting in a learned environment; thus, intergenerational relationships are encouraged and fostered through the differing roles and identities forming and reforming within this age-friendly community. 

## 4. Discussion

This paper has presented an initial overview of the different types of virtual assistant currently available on the market and how these devices can be integrated into existing age-friendly framework(s), coupled with the integration of technology, which to date has not been a focal point of contemporary age-friendly initiatives. This paper sets the scene for initial discussion combining two popular societal domains that are worth exploring further. This review paper uniquely draws together the small volume of literature from the fields of gerontology, gerontechnology, human computer interaction (HCI) and disability. Furthermore, evidence of worldwide ageing populations and the phenomenal developments of technology, in conjunction with the needs of local and national governments, means that alternative solutions are required to address the concerns of citizens ageing-in-place, be it from the standpoint of an older person or a parent/guardian of a dependent person. This paper contributes to—and is at the intersection of—the fields of gerontology, HCI and disability. Consequently, it offers insights into further discussions in the age-friendly and technology [69] domains. 

In the context of older and dependent adults, contemporary evidence illustrates a myriad of opportunities for developers, researchers, health and social care practitioners, older carers and their young dependent children/adults so they can live together, in an environment that is familiar, safe and adaptable to the varying and changing needs of both. 

Despite an increase in evidence, there is still a lack of understanding of the barriers and enablers to the take-up of technology by older adults, their support networks and healthcare practitioners. Given the nature of preventative healthcare technology and the potential influences it has in day-to-day activities, there is the self-perception and assumption that technology is not suitable for carers, marginalized and vulnerable communities, dependent adults and children, commonly based on their identification as being frail and/or lacking experience/understanding. Therefore, there is a need to improve understanding of the importance of planning and prevention at an early stage. Moreover, we need to demonstrate and highlight the benefits of technology in one’s life, family environment and across communities, who may, in turn, want to use technology to enhance their intergenerational experiences and relationships. The intergenerational exchange of knowledge and experiences can be shared and passed on to younger people. At the same time younger people can facilitate a sense of meaning and purpose for older adults. These shared experiences and community involvement can offer and identify specific meanings to all residents, who may have several identities and roles within the family, community, local area and community groups [16]. Burke and Stets [16] purport that a person’s myriad identities are interconnected through their respective behaviour(s), feelings, judgements and sentiments, which are influenced and integrated through identity and society. Identity is associated with one’s role in society or community—this could be through their profession, being a member of a community group (e.g., church, organisation) or network. Each identity has its own characteristics and expectations associated with the respective identity, resulting in one’s expectations being integral in the transformation of powerful stereotypes [16]. 

The authors have discussed the use of virtual/personal assistants such as Alexa, Echo and Google Home. This type of technology offers users across society a variety of options and support in their day-to-day tasks. For example, a British man who has cerebral palsy uses his virtual assistant to ensure he is able to get in and out of bed safely [70]. Connecting the virtual assistant to a light bulb in the bedroom and speaking the correct commands (i.e., switch light on/off) can offer a person enhanced safety (reducing the risk of falling) and independence. This type of support or assistance ensures a user who suffers from a disability, chronic health or life-limited condition the dignity, power and control over his or her own life. While it is still necessary for carers and support networks to assist individuals with deliberating conditions, virtual and personal assistants seem to offer users greater control. 

Ferguson and Damodoran [23,24,26] have discussed and highlighted the needs and requirements of grassroots networks surrounding the issues and concerns of the digital divide, while offering and proposing solutions to local and national governments. Several recommendations include taking a ‘user pull’ approach rather than a top-down approach, to enhance and offer greater opportunities to communities and marginalized communities. The suggested ‘user pull’ approach encompasses 15 characteristics, including individuals who are community-based and trusted, drop-in sessions, user centred practice enabling individuals to choose and set their own learning pathways, no demands or assessments placed on the individual(s), and peer-to-peer learning. While fostering this type of physical space, additional benefits are offered to users, including flexibility, which in turn enables users to try new technologies without experiencing pressure from others (for example, sales/retail assistants). Furthermore, by offering a safe, approachable, flexible and peer-to-peer learning space, users’ fears and anxieties regarding learning new technologies are reduced. Similarly, this type of physical space is paramount for individuals in the homeless community who wish to seek health information and advice [25]. 

Using and engaging with a virtual assistant may facilitate the dependent adult to have an identity and role within their environment, where previously this may not have been the case or may have been very limited. From the standpoint of the carer, this may provide a greater sense of freedom, knowing this virtual assistant has the capabilities to offer their dependent adult or child more confidence to conduct different activities. Moreover, the use and deployment of smart home devices, wearable devices and communication tools such as virtual assistants can offer ageing carers and dependent adults the option to monitor their daily activities, their wellbeing and their quality of life. Additionally, this form of technology can offer inter-generational support, resulting in the perception of the role and identity of a person to be positive, an authority figure role within the family, peer group(s) or community group(s) as a tech-savvy dyad [16]. However, little is known about the use and impact of technologies and the positive benefits of deploying virtual assistants into the lives of ageing carers and dependent adults on a day-to-day basis. This is also the case with the age-friendly framework, and in ascertaining whether one or both actors can age-in-place in their respective communities when faced with the barriers and enablers of being digitally connected or disconnected. 

## 5. Recommendations and Thoughts

Future work should incorporate technology being tested and used in real-life settings, with dependent adults, their carers and support networks. Additional investigations should seek to include health practitioners to examine their perspectives and impact of virtual assistants within their role(s) and identity within the community. 

Concomitantly, the cost-effectiveness of virtual assistants and associated technologies needs to be explored and taken in to account. This would not solely relate to purchasing of technology, but also the integration of technology into new construction projects (e.g., housing) and infrastructure [17,28]. Contemporary research and policy briefings show few or no evidence-based recommendations associated with the cost of installing the Internet, and this should be evaluated for both short-term and long-term adherence, focusing on a cost–benefit analysis to ascertain whether the cost implications outweigh the benefits of integrating the technology into the lives of older adults. For some actors, there is a perception that a piece of technology is a luxury or an unnecessary bill, which in turn may outweigh the benefits and take-up. For users on a low income, whose income may already be stretched, the added necessity of an Internet bill may not be an incentive or motivation to invest, regardless of the potential benefits.

Greater exploration is needed to examine the barriers and enablers of technology associated with existing carers and people diagnosed with diverse disabilities. This work has the potential to ascertain the impact that such technology has or may have on successful ageing and ageing-in-place. Therefore, conducting this examination would reveal myriad perceptions and impacts associated with technology, home automation and the issues associated with this integration and use. 

Across the healthcare sector, services are aiming to be more cost-effective, and technology has the potential to offer alternative solutions (e.g., Skype consultations/appointments). However, the paucity of evidence from the standpoint of the health practitioner demonstrates the need for this area to be explored. Coupled with health provisions, it is necessary to ensure local and national policymakers are informed of contemporary evidence to safeguard and ensure that community and national infrastructure is available to deliver digital solutions to all members of society. It is particularly important for networks and communities at a grassroots level to have an input and voice. Previously, policy briefings have been given at the Northern Irish Assembly [71,72] in Belfast that demonstrate how contemporary research can be used to inform policymakers and community actors who can make a difference at the local government level, and who represent their respective communities. 

In the context of age-friendly environments, the GHC Hope Meadows environment illustrates the positivity and benefits of intergenerational residents residing in one environment. Therefore, the use and deployment of smart home devices, wearable devices, and communication tools offer residents in this type of environment or their respective digital eco-system the option to monitor their health and wellbeing, their daily activities, and also that of their neighbours. 

Contemporary evidence and work surrounding age-friendly frameworks has made great strides [69], yet, there is still a paucity of work on understanding of the impact technology can have on the physical space of a home, shared community or outdoor space. Future work should consider exploring the age-friendly agenda in conjunction with technology, taking on board the suggestions posed by Ferguson and Damodoran [23,24,26] to take a ‘user pull’ approach, furthering previous initiatives and ensuring local governments can support volunteers through the provision of physical space, advice and need in order to support those who are vulnerable, slow adopters or who are homeless. 

## 6. Conclusions

This review paper is significant because it draws on the work from the fields of gerontology, HCI and age-friendly framework(s). Based on the evidence, there is a paucity of current debates focusing attention on the power technology can have within and across age-friendly cities and communities. This is particularly the case for those individuals who are carers in later life to dependent adults. 

Furthermore, this paper brings together a myriad of domains to discuss contemporary issues surrounding individuals and communities of the 21st century society. While there are phenomenal technological developments occurring through artificial intelligence (AI), interaction (e.g., gesture, voice) and engagement, it cannot be ignored that there are still concerns surrounding access and digital participation. 

To alleviate and close the gap of the digital divide requires substantial work relating to and focusing on communication and co-production from all directions of society. This would require groups including local and national governments, education providers, charities, architects, construction, families, and generational cohorts and businesses to collaborate together and move this agenda forward. Circulating strategies are not productive when the needs of the most vulnerable or marginalized communities are not met. Ensuring the infrastructure of a city/town or country is accessible to those wealthiest as well as those who are in marginalized communities is key. Exploring and identifying issues surrounding infrastructure could be useful, such as offering free Wi-Fi on public transport (e.g., train service), and public spaces which in turn allow individuals such as rough sleepers to access an Internet connection while also facilitating them to search specific information (e.g., health). The UK has experienced 10 years of austerity, witnessed across varying regions up and down the country, that has left the neediest and most vulnerable in extremely difficult circumstances. Businesses—be they large or small—have a social responsibility to assist and support local and national initiatives and communities. This too is the responsibility of county councils and government(s). 

All proposed recommendations and future proposals should work towards the culmination of industry, community networks, health practitioners, families, and policymakers to learn and share knowledge, experience, and share ‘lessons learned.’ More importantly, all actors playing a role and part in decision making need to listen to the voices of those directly affected, while also identifying the needs of the people at all socio-economic levels of society. 

The work presented in this paper contributes to the fields of gerontology, gerontechnology, (HCI) and disability, based on the debates associated with the integration of new technologies into the home and/or physical space used by citizens in society with/out a myriad of disabilities. 

There is the potential for virtual/personal assistants to positively impact the lives of carers of dependent adults, children and adults with chronic health and neuro-degenerative conditions. Conducting a co-production approach with a multitude of actors has the potential to move age-friendly framework(s) forward. Taking this kind of approach will ensure all voices are heard, especially the voices of those who will be residing in these environments or cities, by those actors who will primarily be responsible for making the final decision(s). 

## Figures and Tables

**Figure 1 healthcare-07-00049-f001:**
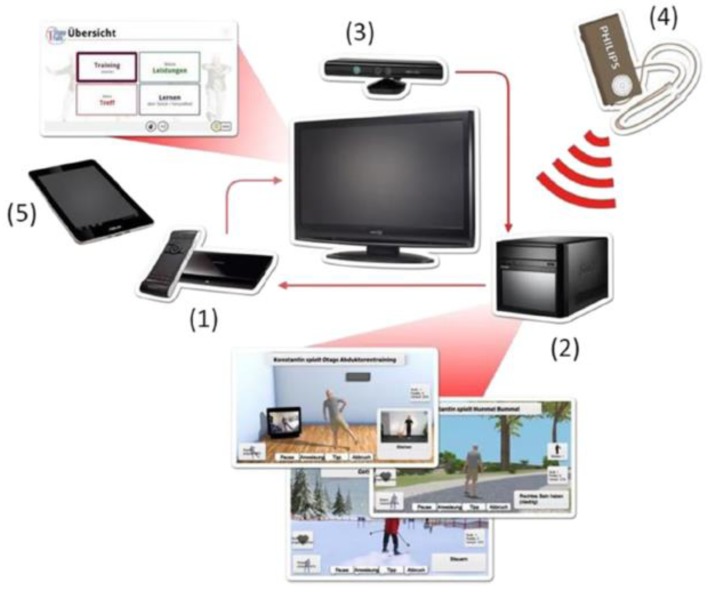
The different technologies integrated into the iStoppFalls ICT-based system. (Permission granted by the Dr Rainer. Wieching—PI, [57].) (**1**) Set top box (iTV), (**2**) mini-PC (exergame), (**3**) Kinect (gesture/voice), (**4**) Senior Mobility Monitor, (**5**) tablet (diary, control).

**Figure 2 healthcare-07-00049-f002:**
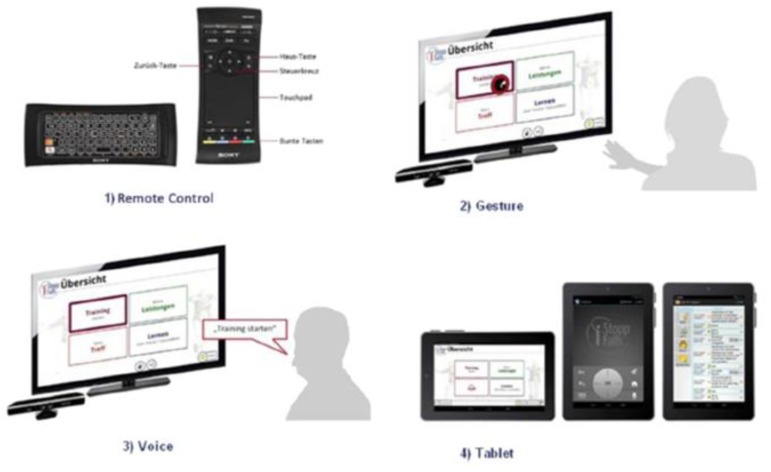
The different modes of interaction available to users of the iStoppFalls ICT-based system. (Permission granted by the Dr Rainer Wieching—PI, [57]). (**1**) Remote control, (**2**) gesture, (**3**) voice or (**4**) tablet.

**Figure 3 healthcare-07-00049-f003:**
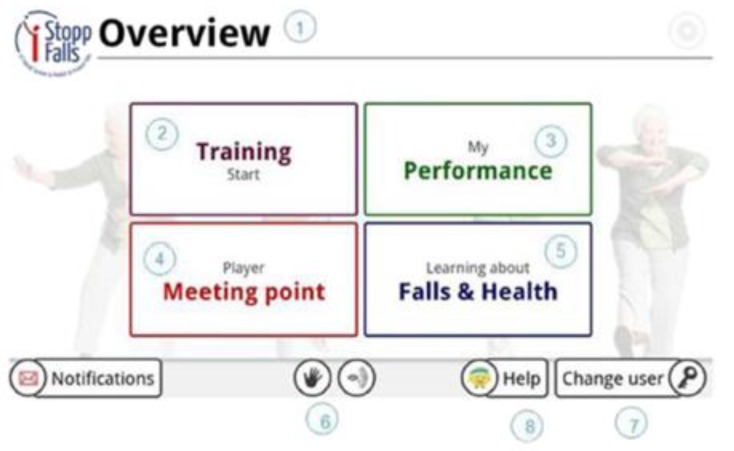
The different menu options available to users in the iStoppFalls ICT-based system. (Permission granted by the Dr Rainer Wieching—PI, [57]). (1). Page header. (2) Training: The area of training. The user can exercise or determine their risk of falling. (3) Performance: The user can view their feedback and results. (4) Meeting Point: The user can communicate with other users who use the system. (5) Falls & Health: The area of learning, educational material and information on fall risks in everyday life, and how to reduce this risk. (6) Gesture and Voice Recognition: Two buttons to activate the gesture and/or voice control. (7) Change User: The user can either log out of the program or start with a different user account. (8) Help: The user can find help in this section for the most common problems and how to use the system.

**Figure 4 healthcare-07-00049-f004:**
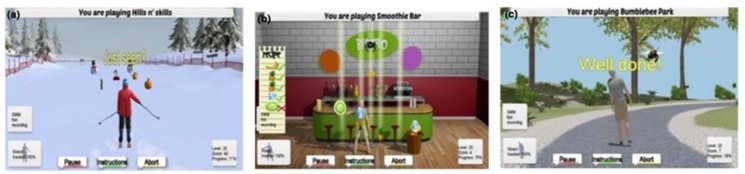
The three purpose-built exergames ((**a**). Hills ‘n’ Skills, (**b**). The Bistro and (**c**). Bumble Bee Park) integrated into the iStoppFalls ICT-based system. (Permission granted by the Dr Rainer Wieching—PI, [57]).

**Figure 5 healthcare-07-00049-f005:**
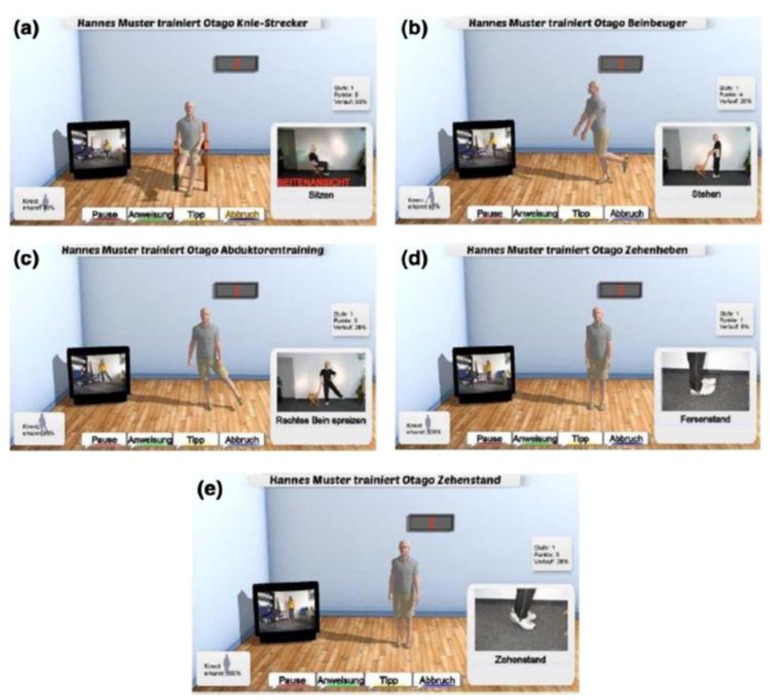
The different Otago exercises integrated into the iStoppFalls ICT-based system. (Permission granted by the Dr Rainer Wieching—PI, [57]). (**a**) Knee extension, (**b**) knee flexion, (**c**) leg abduction, (**d**) toe raises, (**e**) calf raises. There is a demonstration via the icon on the bottom right hand side of the screen. On the right side of the screen, the users are able to see themselves on the television screen. The four buttons at the bottom of the screen (pause, instructions, tips, and abort) can be selected by the users to execute the command.

**Figure 6 healthcare-07-00049-f006:**
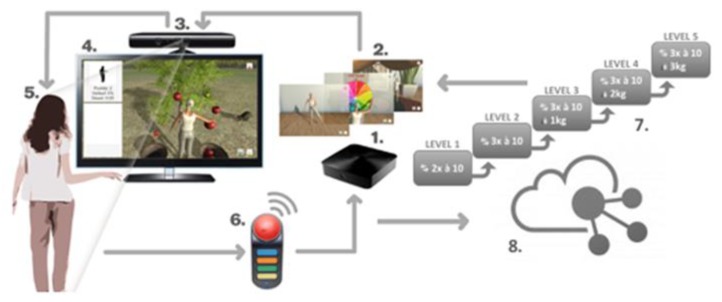
An overview of the MobiAssist ICT-based system and the technological components. (Permission granted by David. Unbehaun [63,64]).

**Figure 7 healthcare-07-00049-f007:**
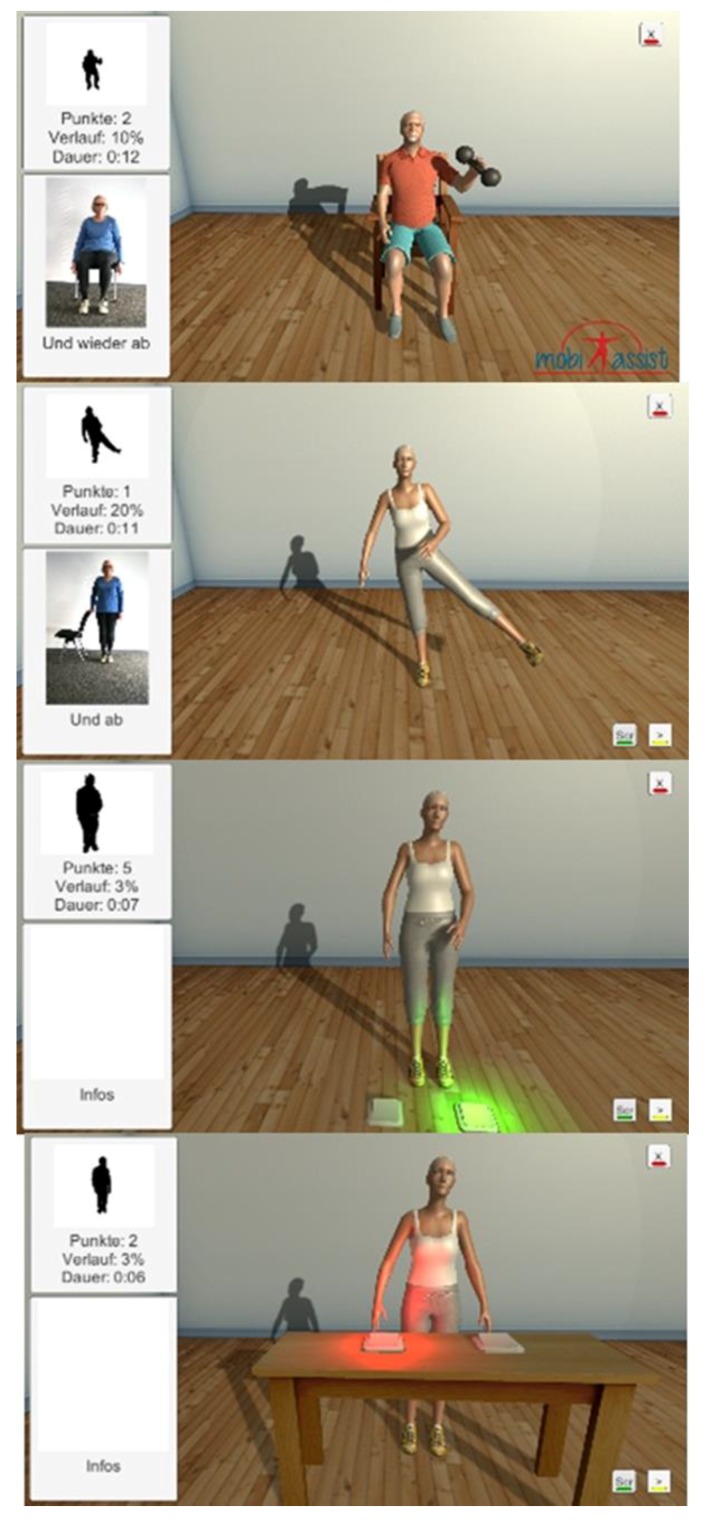
Integrated assessments in the MobiAssist ICT-based System. (Permission granted by David. Unbehaun) [65].

**Figure 8 healthcare-07-00049-f008:**
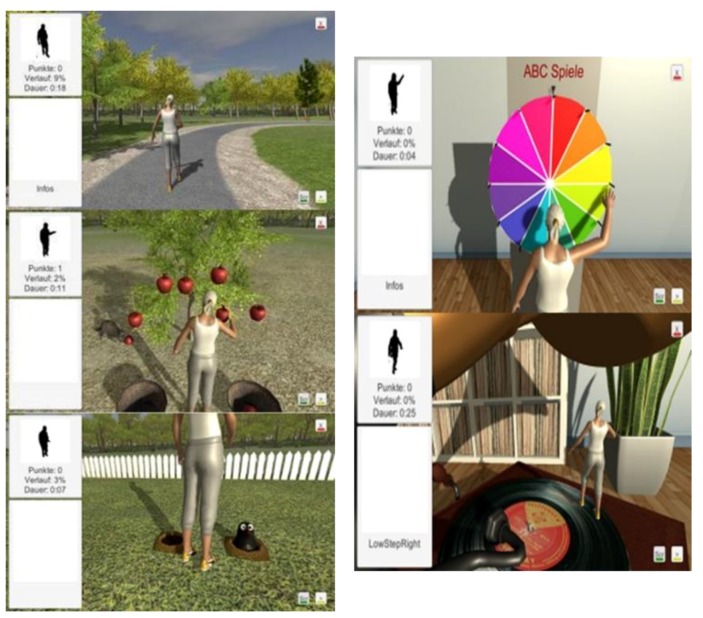
(**Left**) displays the strength and balance Park, Apple and Mole exergames. (**Right**) displays the Wheel of Fortune and Music/walking on the spot exergames. (Permission granted by David. Unbehaun) [64].

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
