# Peer review of "A Review of Age Friendly Virtual Assistive Technologies and their Effect on Daily Living for Carers and Dependent Adults"

_healthcare, 2019, doi:10.3390/healthcare7010049_

Round 1

Reviewer 1 Report

The paper consists initially of literature review, scoping and critical examination of background literature relating to Smart-Home technologies and their potential benefit for older people/ people with disabilities/ carers. It then proceeds with a description of recent tech developments and active research projects in this field, with some critical analysis of the benefits in daily living of these technologies to the above groups. It then concludes with a discussion of potential future developments and conclusion.

There seems to be some confusion as to the thematic thrust of the work. The introduction claims that the work will focus on older carers of dependents with Autism Spectrum Disorder and their relationships with one another and how these can benefit from the technologies assessed, but this is not addressed in the title and I would argue is not particularly dealt with in a meaningful and critical way within the text. The various technologies presented are assessed in relationship to a series of potential benefits but not in any real way in relation to ASD individuals and their carers. Either the work needs significant revision in this regard or (and this would be my suggestion) any specificity to ASD is removed from the text and it is included simply as one of a series of cognitive and physical impairments requiring care. If ASD and those affected by it remains the thrust of the work then there needs to be a lot more analysis specific to those on the spectrum and those (older people) that care for them.

If the ASD route is still to be specifically pursued then there should be more of a section on ASD Friendly places and environment which should reflect on key differences or commonalities with age friendly environments. There is a wealth of literature on environment and ASD now.

The paper would benefit from a bit more rigour methodologically. A short methodology section would be useful, explaining the relationship between the more general literature review section and the more specific case studies. Can you demonstrate that the case study method is the best available? Might you not benefit from defining some key parameters of analysis? How did you select the case studies? Why are these ones chosen and not others. There may even be a way of creating some data of your own through more rigorous analytical methods.

Your sources are very thorough and a lot of information is presented but it does not present itself yet as narratively coherent. The relationship between the different parts must feel more sequential and compelling.

There is quite a lot of grammatical clumsiness and error which needs to be ironed out in any re-draft.

I think a more rigorous organisation and methodology would be a big benefit.

Author Response

On behalf of all the authors, I would like to thank this reviewer for their detailed and constructive feedback relating to our submission.

The paper consists initially of literature review, scoping and critical examination of background literature relating to Smart-Home technologies and their potential benefit for older people/ people with disabilities/ carers. It then proceeds with a description of recent tech developments and active research projects in this field, with some critical analysis of the benefits in daily living of these technologies to the above groups. It then concludes with a discussion of potential future developments and conclusion.

The aim of this paper was not to undertake a literature review. We wanted to

There seems to be some confusion as to the thematic thrust of the work. The introduction claims that the work will focus on older carers of dependents with Autism Spectrum Disorder and their relationships with one another and how these can benefit from the technologies assessed, but this is not addressed in the title and I would argue is not particularly dealt with in a meaningful and critical way within the text. The various technologies presented are assessed in relationship to a series of potential benefits but not in any real way in relation to ASD individuals and their carers. Either the work needs significant revision in this regard or (and this would be my suggestion) any specificity to ASD is removed from the text and it is included simply as one of a series of cognitive and physical impairments requiring care. If ASD and those affected by it remains the thrust of the work then there needs to be a lot more analysis specific to those on the spectrum and those (older people) that care for them.

If the ASD route is still to be specifically pursued then there should be more of a section on ASD Friendly places and environment which should reflect on key differences or commonalities with age friendly environments. There is a wealth of literature on environment and ASD now.

We have taken out the majority of references relating to ASD and we have rewritten the paper to form a more general discussion relating to technology use in Age-friendly homes and disabilities.

The paper would benefit from a bit more rigour methodologically. A short methodology section would be useful, explaining the relationship between the more general literature review section and the more specific case studies. Can you demonstrate that the case study method is the best available? Might you not benefit from defining some key parameters of analysis? How did you select the case studies? Why are these ones chosen and not others. There may even be a way of creating some data of your own through more rigorous analytical methods.

We have added a methods section to the manuscript. This section underpins the thematic scope of the manuscript and places it in context with the field of research.

Your sources are very thorough and a lot of information is presented but it does not present itself yet as narratively coherent. The relationship between the different parts must feel more sequential and compelling.

Given our restructuring we hope our amendments make a more sequential and compelling read – which has been finalised as a link between each section and tied up in the discussion and conclusion sections.

There is quite a lot of grammatical clumsiness and error which needs to be ironed out in any re-draft.

We placed the manuscript through software used by authors who have been those diagnosed with dyslexia to ensure all errors and wording are detected before resubmitting.

I think a more rigorous organisation and methodology would be a big benefit.

We have taken on board the constructive and positive feedback from Reviewers 1 and 2 and we would like to thank both reviewers for their time in reading our submission. Given the significant restructuring and amendments we have amended the references to reflect these updates. The title of the paper has also been amended.

Reviewer 2 Report

This is a review manuscript centred in the use of technology in order to improve life in older and disabilities people.

I found it very informative and interesting, giving light to different technologies and approaches to use.

However, I think that in the whole the manuscript is too long, and sometimes hard to follow. I think it will be more useful if some of the sections (especially to explaining the different technologies) are shortened. The author should make a short explanation of how they work and usability and their advantages.

Author Response

I found it very informative and interesting, giving light to different technologies and approaches to use.

Thank you for your positive comment. We are pleased that you found our manuscript both informative and interesting.

However, I think that in the whole the manuscript is too long, and sometimes hard to follow. I think it will be more useful if some of the sections (especially to explaining the different technologies) are shortened. The author should make a short explanation of how they work and usability and their advantages.

We have shortened the section relating to the different technologies & we have added a short explanation of how they work and the advantages to this. We have also added examples of the positive use of these technologies by people with different disabilities.

Round 2

Reviewer 1 Report

The structure and content of the paper are now fine however I am concerned at the quality of English grammar which is still not sufficient for academic publication and in some cases makes comprehension difficult.

In my original review in relation to english language and style I found it difficult to decide between the advice of 'extensive editing' and moderate changes. I went for moderate changes and now believe this was the wrong choice.

The work requires extensive editing in order to be grammatically coherent. If this is to be done by the authors it requires to be done in detail and painstakingly. It may be you can include an English editor whose contribution can be acknowledged.

In order to illustrate the extent of change I believe is required I attach an edit of the title, abstract and first 100 lines of the paper with suggested changes.

Author Response

Dear Reviewer,

Thank you again for your time in re-reviewing our revised submission. We truely appreciate your time and constructive feedback and we hope the extensive revisions in particular to language meet your expectations. 

Reviewer 2 Report

The authors have improved the manuscript after followed my suggestion. I also see than many improvements have been done following indications of the other reviewer.

Author Response

Dear Reviewer,

Thank you for your time in re-reviewing our submission. We appreciate your time and effort.